# Interaction of the Gut Microbiome and Immunity in Multiple Sclerosis: Impact of Diet and Immune Therapy

**DOI:** 10.3390/ijms241914756

**Published:** 2023-09-29

**Authors:** Sudhir Kumar Yadav, Kouichi Ito, Suhayl Dhib-Jalbut

**Affiliations:** 1Department of Neurology, Rutgers-Robert Wood Johnson Medical School, Piscataway, NJ 08854, USA; yadavsk@rwjms.rutgers.edu (S.K.Y.); itoko@rwjms.rutgers.edu (K.I.); 2Rutgers New Jersey Medical School, Newark, NJ 07101, USA

**Keywords:** central nervous system autoimmunity, multiple sclerosis, gut immunity, gut microbiota, diet, disease-modifying therapies

## Abstract

The bidirectional communication between the gut and central nervous system (CNS) through microbiota is known as the microbiota–gut–brain axis. The brain, through the enteric neural innervation and the vagus nerve, influences the gut physiological activities (motility, mucin, and peptide secretion), as well as the development of the mucosal immune system. Conversely, the gut can influence the CNS via intestinal microbiota, its metabolites, and gut-homing immune cells. Growing evidence suggests that gut immunity is critically involved in gut–brain communication during health and diseases, including multiple sclerosis (MS). The gut microbiota can influence the development and function of gut immunity, and conversely, the innate and adaptive mucosal immunity can influence microbiota composition. Gut and systemic immunity, along with gut microbiota, are perturbed in MS. Diet and disease-modifying therapies (DMTs) can affect the composition of the gut microbial community, leading to changes in gut and peripheral immunity, which ultimately affects MS. A high-fat diet is highly associated with gut dysbiosis-mediated inflammation and intestinal permeability, while a high-fiber diet/short-chain fatty acids (SCFAs) can promote the development of Foxp3 Tregs and improvement in intestinal barrier function, which subsequently suppress CNS autoimmunity in the animal model of MS (experimental autoimmune encephalomyelitis or EAE). This review will address the role of gut immunity and its modulation by diet and DMTs via gut microbiota during MS pathophysiology.

## 1. Introduction

Multiple sclerosis (MS) is a chronic autoimmune inflammatory disease of the central nervous system (CNS). MS is characterized by peripheral immune dysregulation and immune cell infiltration into the CNS, leading to demyelination and axonal damage, and, ultimately, neurodegeneration. It is suggested that CD4+Th1/Th17, B cells, and CD8+T cells play an important role during the initial inflammatory phase of MS [1]. In addition, glial cells such as microglia and astrocytes play critical roles during the neurodegenerative phase of MS. Both genetic and environmental factors contribute to immune dysregulation during MS onset and progression [2]. Earlier, we reported that the gut microbiota interacts with MS susceptibility genes to break immune tolerance to myelin antigens and leads to the development of experimental autoimmune encephalomyelitis (EAE) in 3A6 TCR/DR2a Tg mice [3]. So far, nine classes of disease-modifying therapies (DMTs), including interferons, glatiramer acetate, teriflunomide, sphingosine 1-phosphate receptor modulators, fumarates, cladribine, and monoclonal antibodies, have been approved by regulatory authorities to treat MS. These DMTs have been helpful in reducing clinical relapses and gadolinium-enhancing lesions on brain magnetic resonance imaging (MRI) [4].

The CNS influences gut physiological (gut motility and secretion) and immunological (development and functioning of the mucosal immune system) functions, while the gut influences CNS development and function via the intestinal microbiota and its metabolites, gut-homing immune cells, gut hormones, and the vagus nerve [5,6]. Such bidirectional communication between the gut and the CNS is commonly known as the gut–brain axis and could be a potential target for therapeutic intervention. Accumulating evidence indicates that brain-resident and gut-resident immune cells are critically involved in orchestrating gut–brain axis communication. The gut microbiota can influence the development and function of the intestinal immune system, and, conversely, the innate and adaptive immune system can influence microbiota composition [7]. Recently, we demonstrated in an MS animal model that perturbed gut immune homeostasis is associated with EAE development [8]. Similarly, immune dysregulation has been demonstrated in MS with excessive Th17 cell expansion in the intestine [9]. These reports suggest that gut immunity has a significant role in the immunopathogenesis of MS. Since diet and DMTs can influence gut immunity by affecting the gut microbial community [10], we hereby review their roles in modulating MS pathophysiology via gut immunity. 

## 2. Role of Gut–Immune Axis in MS Pathogenesis

In a spontaneous EAE mouse model, we recently showed that CNS autoimmunity is triggered by altered gut and peripheral immunity due to gut dysbiosis [3,8]. Similarly, altered gut microbiota and gut immunity were found to be correlated with high disease activity in MS patients [9]. In addition, the transplantation of stool samples from MS patients into germ-free mice can increase the severity and incidence of EAE in the transplanted mice [11,12]. These reports suggest that gut dysbiosis may trigger the initiation and progression of CNS autoimmunity by promoting immune dysregulation in the gut (Figure 1).

Humanized gnotobiotic EAE mouse models contributed significantly to the understanding of the role of gut microbiota in MS pathogenesis [11,12]. However, the lack of standardized protocols, interspecies variation between mouse and human, and the type of fecal sample used (RRMS vs. SPMS vs. PPMS) in creating humanized gnotobiotic mouse models can lead to translatability issues. Interspecies parameters that can affect fecal transplantation are differences in human and mouse gut anatomy, digestion process and metabolic rate (quicker in mouse), and mouse genetic and immunological background [13]. In addition, another important limitation is the lack of a clear understanding of the cause–effect relationship between MS and intestinal microbiota dysbiosis. 

Although they appear similar at the phyla level, human and murine gut floras have key discrepancies in the microbial composition and abundance. For example, a higher *Firmicutes*/*Bacterioidetes* ratio is observed in humans compared to mice. Further, the phylum *Bacteroidetes* mainly consists of the S24-7 family, and *Firmicutes* consists of *Clostridiales* in mice. But *Bacteroidetes* mainly consists of *Bacteroidaceae, Prevotellaceae*, and *Firmicutes* of the *Ruminococcaceae* family in humans [14]. Therefore, the results from EAE models should be interpreted with caution while studying the role of the gut microbiota in MS.

### 2.1. MS-Associated Gut Microbiota 

The gut microbiome of treatment-naïve early-stage MS patients of different ethnicities (Caucasian, Hispanic, and African American) have increased relative abundances of *Clostridia* species compared to ethnicity-matched controls. However, other taxa showed significant differences among different ethnicities [15]. In both progressive MS and RRMS, *Clostridium bolteae, Ruthenibacterium lactatiformans,* and *Akkermansia muciniphila* were increased, while *Blautia wexlerae, Dorea formicigenerans*, and *Erysipelotrichaceae CCMM* were decreased. However, increased *Enterobacteriaceae* and *Clostridium* g24 FCEY and decreased *Blautia* and *Agathobaculum* were unique to progressive MS. Interestingly, several *Clostridium* species were associated with higher EDSS and fatigue scores [16]. A certain gut microbiota composition may be associated with subsequent MS relapse, especially in pediatric MS. Notably, *Fusobacteria* depletion was associated with pediatric MS relapse [17]. Future studies should focus on identifying more bacterial species involved in MS relapse. Except for increased *A. muciniphila*, the bacterial taxa involved in the development of gut dysbiosis among individuals with MS varied among studies [11,12,16,18]. These variations may be due to interindividual differences in the baseline microbial composition caused by host genetic factors, long-term dietary habits, environmental exposures related to race/ethnicity, and/or geographical location [15,19,20]. It is unknown whether increased *A. muciniphila* contribute to MS pathogenesis or are a consequence of the disease. Interestingly, a recent report suggests a link between increased *Akkermansia* and lower disability, suggesting that *Akkermansia* may have a beneficial role [16]. It is important that future studies explore the disease-specific roles and mode of action of MS-associated gut bacteria.

### 2.2. MS-Associated Dysregulation of Gut Immunity 

Dysregulated gut and peripheral immunity are consequences of gut dysbiosis (change in gut microbiota). Many reports suggest a link between MS pathogenicity and gut immunity. For example, Th17 cells are now widely accepted to be key players in MS pathogenesis, and an increased frequency of intestinal Th17 cells correlates with high disease activity and altered gut microbiota in MS patients [9]. Also, disruption of the intestinal barrier and increased permeability, often referred to as “leaky gut”, are involved in the pathogenesis of autoimmune diseases [21]. Interestingly, intestinal barrier dysfunction develops at the onset of EAE, which is associated with Th17 cell infiltration in the small intestine as well as an increase in intestinal permeability [22]. Distinct signals from gut microorganisms coordinately activate myelin oligodendrocyte glycoprotein (MOG)-specific Th17 cells in the small intestine. Germ-free mice colonized with two bacteria from the small intestine, a strain from the *Erysipelotrichaceae* family and *Lactobacillus reuteri*, develop more severe EAE compared to germ-free or monocolonized mice. The strain from *Erysipelotrichaceae* acts as an adjuvant to enhance the Th17 cells’ response, while *Lactobacillus reuteri* possesses peptides that potentially mimic MOG. Therefore, the synergistic effects of these microorganisms may be involved in the pathogenicity of MS [23]. MOG-specific Th17 cells infiltrate the colonic lamina propria prior to the development of neurological symptoms in active and adaptive transfer EAE models and alter gut microbiota composition. Disrupting Th17 cell trafficking to the large intestine significantly attenuates EAE [24]. Likewise, we have demonstrated the infiltration of myelin basic protein (MBP)-specific Th17 cells as well as the recruitment of neutrophils in the colon of a spontaneous EAE mouse model [8]. Of note, neutrophils can promote Th17 cell differentiation by neutrophil extracellular traps (NET) and their histones via the Toll-like receptor (TLR) pathway [25]. These reports suggest that the gut may be a location for the differentiation of encephalitogenic Th17 cells in the periphery, with the microbiota playing an important role in their differentiation, activation, and migration to the CNS. It was hypothesized that this could be the result of cross-reactivity between bacterial antigens and endogenous CNS antigens (molecular mimicry) or bystander activation [26]. Recently, we discovered a variant of surface layer protein A (SLPA) in a subtype of *Clostridioides difficile* (strain DJNS06-36), which can activate MBP89-98-reactive T cells. SLPA contains an amino acid sequence that resembles immunodominant myelin basic protein 89–98. Importantly, active immunization with SLPA activates MBP-specific T cells and induces EAE in MBP-TCR/DR2a Tg mice. This study suggests that the encephalitogenic mimotope of MBP of gut bacteria can activate autoreactive myelin-specific T cells and trigger CNS autoimmunity [27]. In summary, MS-associated gut bacterial species have functional effects on the immune system that can potentially modulate MS pathogenesis (Table 1).

In addition, in severe cases of MS, commensal-specific gut IgA responses are drastically reduced, with a simultaneous increase in serum IgG responses against IgA-unbound bacteria compared to controls [31]. Further, the mobilization of IgA+ Plasma blast and/or plasma cells from the gut to the CNS can significantly suppress neuroinflammation [32]. In healthy individuals, C-C chemokine receptor type 9 (CCR9)+ memory T cells exhibited a regulatory profile characterized by both the expression of C-MAF and the production of IL-4 and IL-10. However, in CCR9+ memory T cells, the expression of RORγt was specifically upregulated, and the production of IL-17A and IFN-γ was high in patients with secondary progressive MS (SPMS) compared to healthy controls, indicating the loss of regulatory function [33]. These animal and clinical studies suggest that an imbalance in the gut microbiome between anti-inflammatory and pro-inflammatory bacteria may promote immune dysregulation and increase the risk of MS.

### 2.3. Dysregulated Gut Immunity May Promote MS Relapses 

Gut dysbiosis-mediated intestinal inflammation could be a risk factor for disease exacerbation in MS. Recently, we showed that CNS autoimmunity is associated with gut inflammation, which is probably triggered by a reduction in the enteric bacteria involved in the development of regulatory immune cells. Also, we observed the gut infiltration of Th1 and Th17 cells as well as the recruitment of neutrophils during the development of spontaneous EAE [8]. Inflammatory cytokines like TNF-α, IFN-γ, and IL-1β produced by immune cells during gut inflammation disrupt tight junction proteins, resulting in increased intestinal permeability [34,35]. Notably, intestinal permeability is increased in patients with MS compared with healthy donors [36]. This could be one of the risk factors involved in disease exacerbation. Intestinal permeability induces the translocation of microbial components into the systemic circulation, which could break peripheral immune tolerance [3]. Animal studies also show that increased intestinal permeability exacerbates EAE and promotes disease progression [22]. Further, microbial dysbiosis can affect the production of bacterial metabolites that promote or suppress CNS autoimmunity. For example, microbial dysbiosis reduces the production of short-chain fatty acids (SCFAs), which are important metabolites for the development of forkhead box P3 (Foxp3) Tregs and for the maintenance of immune homeostasis [37,38]. Therefore, the gut dysbiosis-mediated reduction of SCFAs could be a risk factor in MS. Finally, intestinal regulatory and pathogenic immune cells can migrate from the intestine to the CNS [39,40]. Therefore, gut dysbiosis may reduce the migration of regulatory cells and increase the migration of pathogenic immune cells to the CNS. Collectively, these studies suggest that dysregulated gut immunity may promote MS disease activity.

### 2.4. Contribution of Gut Microbiota to CNS Pathology in MS

MS pathology involves inflammatory and neurodegenerative processes. Neuroinflammation is predominant in the early stages of MS, which is mediated by immune cells, whereas neurodegeneration is dominant in the later stages of MS and is mainly driven by microglia and astrocytes, whose activity can be modulated by the gut microbiota [41]. An immature microglia phenotype with diminished immune function and enhanced proliferation and survival observed in germ free mice compared to specific-pathogen-free (SPF) mice, point to the role of microbiota in microglia development and function [42]. Further, a subtype of reactive astrocytes (A1 astrocytes) is induced by activated neuroinflammatory microglia by secreting IL-1α, TNF-α, and C1q [43]. In addition, transforming growth factor-α (TGF-α) and vascular endothelial growth factor-B (VEGF-B), produced by microglia, regulate the pathogenic activities of astrocytes in EAE. Microglia-derived TGF-α limits pathogenic activities of astrocytes and acts via the ErbB1 receptor during EAE. Conversely, microglial-derived VEGF-B activates fms-related receptor tyrosine kinase-1 (FLT-1) signaling in astrocytes to worsen EAE. Interestingly, the metabolites produced by the commensal flora from dietary tryptophan control microglial activation and TGF-α and VEGF-B production by the aryl hydrocarbon receptor [44]. Similarly, neurotoxicity in MS is induced via the microbially derived metabolites, phenol and indole, produced by the tryptophan and phenylalanine catabolism [45]. During homeostatic conditions, a subset of astrocytes expresses the lysosomal-associated membrane protein 1 (LAMP1) and the TNF-related apoptosis-inducing ligand (TRAIL). The TRAIL expression in the astrocytes is promoted by interferon-γ (IFN-γ) produced by meningeal natural killer (NK) cells. Notably, IFN-γ expression in the NK cells are modulated by the gut microbiome. These LAMP1+TRAIL+ astrocytes limit neuroinflammation in the CNS by inducing T cell apoptosis through TRAIL–DR5 signaling. However, during inflammation, TRAIL expression in astrocytes is suppressed by molecules produced by T cells and microglia [46]. In addition, the gut microbiota may affect CNS pathology by producing toxins. For example, *Clostridium perfringens*, which can produce epsilon toxin (ETX), were found in the majority of RRMS patients. Importantly, ETX can disrupt the blood–brain barrier to promote multifocal lesions in the brain and spinal cord of the EAE model, resembling MS lesion pathology [47]. These reports suggest that the gut microbiota can modulate glial cell phenotypes and the blood–brain barrier to promote CNS pathology in MS.

## 3. Mechanisms of the Gut–Immune-Axis-Mediated Effect in the CNS

The gut–immune axis affects the CNS during health and disease by several mechanisms (Figure 2). First, microbial metabolites and endogenous components from the gut microbiota can pass to the circulation and affect peripheral immunity during health and disease. The most notable microbial metabolites are SCFAs like acetic acid, butyric acid, and propionic acid. SCFAs are produced by the fermentation of dietary fiber in the colon by anaerobic gut bacteria [48,49,50]. In RRMS patients, the fecal levels of acetate, propionate, and butyrate are significantly lower compared to HC [51]. Interestingly, female RRMS patients showed significantly reduced fecal SCFA concentrations compared to male RRMS patients, possibly contributing to the higher female susceptibility to MS [52]. SCFAs are known to suppresses autoimmunity through immunomodulatory effects, which have been attributed partially to the epigenetic modulation of immune cells via the inhibition of the histone deacetylase (HDAC) enzyme. One of the downstream effects is enhanced regulatory T cells (Tregs), which suppress autoimmunity [53]. Indeed, the number of CD4+ CD25+ FOXP3+ Tregs are decreased in MS patients and have impaired suppressive capacity [54,55]. The SCFA pentanoate suppresses autoimmunity by inducing IL-10 production in lymphocytes and reprogramming their metabolic activity towards elevated glucose oxidation. In addition, pentanoate-induced regulatory B cells suppress autoimmune pathology in colitis and MS animal models [50]. Also, tryptophan and its metabolites are emerging as important modulators of mucosal and CNS immunity via the aryl hydrocarbon receptor (AHR). Tryptophan metabolites like kynurenine, kynurenic acid, anthranilic acid, quinolinate, indole-3-acetic acid, indoxyl-3-sulfate, indole-3-propionic acid, and indole-3-aldehyde are produced by the action of gut microbiota on dietary tryptophan [56]. Tryptophan metabolites protect against increased gut permeability through the aryl hydrocarbon receptor by maintaining the apical junctional complex and its regulatory proteins (myosin IIA and ezrin) [57]. In addition, tryptophan metabolites also signal through the AHR in astrocytes and reduce CNS autoimmunity via the Suppressor of Cytokine Signaling 2 (SOCS2)-mediated inhibition of NF-κB-driven inflammation [58]. Interestingly, the tryptophan metabolites kynurenine, kynurenic acid, anthranilic acid, and quinolinate are low in MS serum, indicating a possible role in MS pathogenesis [58]. Further, lower serum tryptophan and indole lactate (tryptophan metabolite) are associated with pediatric MS risk and disease course [59]. Contrary to previous reports, kynurenic acid has been shown to promote the accumulation of Th17-inducing GPR35+ Ly6C+ macrophages in the small intestine of EAE mice before disease induction. *Sporosarcina pasteurii*, *Staphylococcus lentus*, *Pseudoxanthomonas mexicana*, and *Sphingomonas* were identified as potential species involved in kynurenic acid production [60]. Therefore, tryptophan-metabolizing gut bacteria and the metabolic end-product will modulate the effect on CNS autoimmunity. Among bacterial components, polysaccharide-A (PSA) and LPS are well investigated. PSA is a capsular polysaccharide produced by a Gram-negative symbiont, *Bacteroides fragilis,* in the colon. PSA from human gut bacteria *Bacteroides fragilis* protects against EAE by inducing IL-10-producing FoxP3+ Treg cells [61]. Further, PSA promotes human CD39+Foxp3+Treg cells and Treg function [62]. PSA suppresses EAE by the expansion of CD4+Treg. Further, CD39, which is an ectonucleotidase, promotes the accumulation of CD39+CD4+ Tregs in the CNS to suppress autoimmunity [63]. Interestingly, *Bacteroides fragilis* is significantly depleted in pediatric MS patients, suggesting its possible role in MS pathogenesis [64]. In addition, gut dysbiosis during CNS autoimmunity increases the serum level of lipopolysaccharide (LPS), which promotes the loss of peripheral immune tolerance [3,37]. LPS injection has been shown to exacerbate EAE [65]. Further, systemic LPS can activate microglia to increase pro-inflammatory factors in the CNS, consequently contributing to neurodegeneration [66]. In summary, it can be concluded that microbiota-derived metabolites and components will either promote or suppress CNS autoimmunity.

Second, in homeostatic and disease conditions, CNS function and behavior can be affected by resident innate and adaptive immune cell populations [67,68,69]. Interestingly, many of these immune cells are derived from the periphery and their development can be modulated by gut microbiota [70,71]. Depending on the composition of the gut microbiota, the latter can promote or suppress neuroinflammation and/or demyelination via the development of Th1/Th17 or Treg cells, respectively [72,73]. In fact, the dysregulation of TGF-beta/Smad7 signaling has been reported in the intestine of MS patients. This favors an inflammatory phenotype in intestinal CD4+ T cells and leads to migration to the CNS to cause autoimmunity [40]. Further, the gut-homing of myelin-specific Th17 cells is required for disease induction in the adoptive transfer EAE model, suggesting a role for gut microbiota/gut environment in the differentiation of encephalitogenic Th17 cells [24]. Conversely, gut-microbiota-specific IgA+ B cells can traffic to the CNS in active MS and dampen excessive inflammation [32,74]. Therefore, the gut microbiota can affect the CNS during health and disease by affecting immune cell homing and migration pathways.

Third, the vagus nerve and enteric nervous system (ENS) innervate the intestinal wall and play a significant role in bidirectional communication between the gut and the CNS. The ENS can function independently of the vagus nerve to control gut physiology like motility and secretion. The ENS is made of myenteric (Auerbach’s) and submucosal (Meissner’s) plexuses, which are connected to CNS via the vagus nerve [75]. The gut wall is enervated with both afferent and efferent fibers of the vagus nerve and its activity can be modulated by the diffusion of bacterial components (like LPS) and metabolites (like SCFAs) or hormonal signals from specialized enteroendocrine cells (EECs), which are capable of sensing luminal bacterial content [76]. Only a few studies have explored the role of the ENS in CNS autoimmunity. For example, in a B-cell- and antibody-dependent mouse EAE model, the degeneration of the myenteric plexus with gliosis and axonal loss caused a decrease in intestinal motility before the onset of EAE had been reported. Interestingly, gliosis and ENS degeneration were also detected in resected colon from MS patients. Further, both EAE mice and MS patients have serum autoantibodies against antigens derived from enteric neurons and/or glia [77]. Also, altered gastrointestinal motility in EAE models was reported due to autoantibodies targeting the ENS [78]. These reports suggest that the ENS may be involved in the pathogenesis of MS, which warrants further studies.

Lastly, gut hormones can also affect the CNS. It is important to note that gut-microbiota-derived LPS and SCFAs modulate the production and release of gut hormones. Cholecystokinin (CCK), ghrelin, peptide YY, glucagon-like peptide-1 (GLP-1), and 5-hydroxytryptamine (5-HT) or serotonin are some of the important gut hormones produced by EEC in the gastrointestinal tract [79]. GLP-1 potentially suppresses neuroinflammation since it attenuates LPS-induced inflammatory responses in microglia [80]. In fact, Semaglutide, a novel glucagon-like peptide-1 agonist, has been shown to suppress EAE in rats [80]. Most of the body’s serotonin is secreted by EEC. Fluoxetine, a serotonin reuptake inhibitor, and 5-HT suppress Th17-immune responses in multiple sclerosis (MS). Such an effect is facilitated by the activation of 5-HT2B receptors, which reduces IL-17, IFN-γ, and GM-CSF production in MS [81]. Therefore, gut hormones may play an important role in MS pathogenesis.

## 4. Effect of Diet on the Gut–Immune Axis in MS

### 4.1. High-Fiber Diet

Dietary fiber is an important food component involved in the maintenance of health, which can be mediated by microbiota-dependent or independent mechanisms [82]. Dietary fiber is an edible plant carbohydrate polymer that is resistant to digestion and absorption in the small intestine, with complete or partial fermentation in the large intestine. Dietary fiber includes polysaccharides (cellulose, hemicellulose, and pectin), oligosaccharides, lignin, and related plant molecules [83]. Bacterial species from *Bifidobacterium, Prevotella*, and *Bacteroides* are the main degraders of dietary fiber to SCFAs such as butyrate, propionate, and acetate [84]. Dietary fiber and SCFAs can suppress an immune response through G protein-coupled receptors or acting as histone deacetylase inhibitors [85]. SCFAs are critical to maintain intestinal homeostasis and immunity by inducing IL-22 production in CD4+ T cells and innate lymphoid cells 3 (ILC3) [86]. Further, SCFAs drive monocyte-to-macrophage differentiation via histone deacetylase 3 (HDAC3) inhibition. Interestingly, these macrophages have enhanced antimicrobial activity due to a shift in metabolism, a reduction in mammalian target of rapamycin (mTOR) kinase activity, and increased LC3-associated host defense [87]. Therefore, SCFAs can prevent gut dysbiosis by inhibiting the overgrowth of pathogenic bacteria. Importantly, dietary fiber and SCFAs have been shown to inhibit CNS autoimmunity in vivo through the differentiation of Tregs and the inhibition of pathogenic Th cells [88,89]. Tregs differentiation is mediated by the induction of tolerogenic dendritic cells (DC) and/or the enhanced acetylation of the Foxp3 gene by SCFAs through the inhibition of histone deacetylase activity [38,90]. In addition, SCFAs can affect immune cell metabolism and T helper cell differentiation through mTOR signaling [91]. 

Besides immune modulation, SCFAs are the preferred energy source for intestinal epithelial cells and maintain a healthy intestinal epithelial layer by promoting epithelial cell proliferation and turnover [92]. Further, dietary fiber and SCFAs can prevent autoimmunity by promoting intestinal barrier function and preventing the migration of intestinal lymphocytes to extraintestinal tissues [93,94]. The SCFAs produced by the microbiota are important to suppress inflammatory microglia, which partially depend on epigenetic modifications [95]. In addition, butyrate treatment has been shown to suppress demyelination and enhance remyelination by modulating oligodendrocytes directly [96]. SCFAs maintain the blood–brain barrier (BBB) and blood–cerebrospinal fluid (CSF) barrier through the increased expression of tight junction (TJ) proteins as well as cytoskeleton rearrangement in endothelial cells [97,98,99]. SCFA diets alleviate cognitive and spatial memory deficits by enhanced astrocyte–neuron metabolic coupling, leading to reduced oxidative damage [100]. Therefore, dietary fiber and its metabolite SCFAs are important for immune and nervous system homeostasis, thereby helpful in suppressing autoimmunity and neurodegeneration.

MS is highly associated with gut dysbiosis, characterized by the loss of bacterial taxa involved in the fermentation of dietary fiber and the production of SCFAs [8,101]. Since SCFAs can promote the differentiation of Tregs, a decrease in SCFA-producing bacteria in the gut would be a risk factor for MS. Indeed, the oral intake of propionic acid promotes a sustained increase in functionally competent Treg cells, while Th1 and Th17 cells are significantly decreased in MS patients [102]. Similarly, a high-vegetable/low-protein diet (HV/LP diet), which is rich in dietary fiber, is associated with reduced relapse rate and less disability due to microbial composition change and the induction of IL10-producing monocytes and Tregs [103]. Also, animal studies have indicated that feeding a high-fiber diet or diet rich in SCFAs ameliorated EAE through the increased differentiation of Tregs and the downregulation of Th1 cells [50,89]. Important ongoing or completed studies addressing the effects of different diets on MS are summarized in Table 2. Dietary fiber and SCFAs also affect neurodegenerative diseases. For example, fecal SCFA levels are lower in Alzheimer’s disease (AD) patients, and its supplementation can potentially provide therapeutic benefits for AD since they can prevent the formation of neurotoxic Aβ aggregates and Aβ-induced microglia activation [104,105,106]. Thus, an increase in the consumption of a high-fiber diet could be beneficial for neurodegenerative diseases including MS. Recently, a high-fiber supplement was shown to efficiently reduce gut dysbiosis and promote the growth of SCFA-producing gut bacteria in humans [107]. The effect of the high-fiber supplement in MS gut dysbiosis and disease induction in EAE is currently being investigated in our laboratory.

### 4.2. Western Diet/High-Fat Diet (HFD)

Western diets/HFDs can impact gut–brain communications by altering the gut microbiota and intestinal permeability. Commonly, the consumption of a HFD leads to an increase in *Firmicutes* and a decrease in *Bacteroidetes*, which are associated with obesity and chronic diseases [110]. Interestingly, obesity is associated with higher disease activity and poorer outcome in newly diagnosed MS patients [111]. The consumption of a HFD led to unfavorable changes in fecal and plasma metabolite and plasma factors, resulting in poor long-term health outcomes [112]. In addition, HFDs have unfavorable effects due to the dysregulation of gut immunity and barrier function [113,114]. HFDs have been shown to induce neuroinflammation, oxidative stress, and neuronal death in the brain cortex and hippocampus of mice [115]. Further, a HFD exacerbated neuroinflammation in an animal model of MS by promoting microglial activation and T cell infiltration [116]. As the neurotrophic factor BDNF plays a key role in brain function, HFD-induced neuroinflammation suppresses brain-derived neurotrophic factor (BDNF)-related pathways [117]. Similarly, saturated fat increases the risk of relapse in pediatric MS [118]. However, polyunsaturated fatty acids (PUFAs) such as omega-3 fatty acids have been shown to reduce disease severity in an MS mouse model. This beneficial effect is due to the reduced polarization of naïve T cells toward proinflammatory Th1 and Th17 phenotypes [119]. In fact, a higher intake of omega-3 supplementation reduces the risk of MS in patients with clinically isolated syndrome [120]. In addition, omega-3 supplementation has been associated with decreased relapse rate and inflammatory markers and improved quality of life in MS patients [121]. Similarly, omega-6 supplementation decreases pro-inflammatory monocyte number and function, with a simultaneous increase in anti-inflammatory monocyte subsets and functions in MS patients [122]. A HFD increases the risk of neurodegenerative diseases like Alzheimer’s and Parkinson’s disease [123,124], perhaps in connection with HFD-induced gut dysbiosis [125]. 

## 5. Effect of MS Therapeutics on the Gut–Immune Axis in MS

Disease-modifying therapy (DMT)-induced changes in gut microbiota may contribute toward their therapeutic effects (Table 3). For example, by promoting anti-inflammatory gut bacteria, DMTs can enhance Tregs or the production of regulatory cytokines in the gut and the circulation. Recently, the gut microbiomes of 576 MS patients and 1152 genetically unrelated household healthy controls (HHC) were studied by the International Multiple Sclerosis Microbiome Study (iMSMS). In this study, differences in β-diversity were observed when patients within each treatment group were compared to their corresponding HHC. However, β-diversity did not differ between the treated and untreated RRMS patients except for the IFN-β treated group [10]. Similarly, in another study, the overall composition of the microbiota did not differ significantly between treated and untreated MS patients. However, specific bacteria were found to be linked to each DMT [16]. Overall, DMT-induced changes in microbial populations may impact biological and metabolic pathways. For example, glatiramer acetate (GA) and dimethyl fumarate (DMF) mainly upregulate retinol (vitamin A) metabolism, which is known to promote the differentiation of Tregs and suppress the reprogramming of Treg to Th17 cells during intestinal inflammation [126,127]. In addition, both therapies decreased methane metabolism. An increased relative abundance of proinflammatory methane-producing bacteria, *Methanobrevibacter*, and exhaled methane was reported in MS patients compared with healthy controls [18,126]. Therefore, by decreasing methane-producing bacteria, GA and DMF have beneficial therapeutic effects. Although certain DMT-induced changes in the gut microbiota have been identified, no specific microbiota DMT signatures are currently established. This can be explained by two factors: interindividual differences in base microbiota and use of previous DMTs. First, the baseline microbial compositions have interindividual differences caused by host genetic factors, dietary habits, environmental exposures, and/or geographical location [15,19,128]. Therefore, different study cohorts might have different microbiota compositions after DMT. The effects of individual DMTs on bacterial composition are summarized in Table 3. Second, treatment guidelines do not provide guidance on how to select or sequence DMTs, which leads to switches or discontinuations of initial DMTs (~70% of patients) [129]. Therefore, it will be hard to predict a gut microbiota signature for a specific DMT. It is also unknown how the gut microbiota modulates the response to DMT. We hypothesize that the gut microbiota can affect DMT actions by affecting its metabolism via the production of enzymes that degrade or activate drugs [130]. In addition, the gut microbiota can produce some metabolites that could compete with DMTs or affect the level of metabolite transporters, which may affect its bioavailability. Therefore, gut microbiota composition may contribute to the efficacy of DMTs. 

## 6. Conclusions

Emerging evidence suggests important roles for the gut microbiome and gut immunity in human health and disease. MS is associated with perturbed gut microbiomes and immunity, supporting the role of the gut microbiome in the initiation and progression of MS. However, comprehensive knowledge about the functional effects of MS-associated bacteria is lacking at present. It is important that future studies explore the functional effects and mode of action of MS-associated gut bacteria, which would be helpful in designing therapeutic strategies. A high-fiber diet and disease-modifying therapies (DMTs) can have beneficial effects for MS by shifting the composition of the gut microbial community to an anti-inflammatory phenotype, leading to the differentiation of regulatory immune cells and an improvement in intestinal barrier function. However, it is not known how the gut microbiota modulates DMTs; the role played by baseline microbiomes in determining the response to DMTs is also unknown. Studies in this direction could help personalize the choice of DMTs, since the microbiome is varied among different MS populations and disease states.

## Figures and Tables

**Figure 1 ijms-24-14756-f001:**
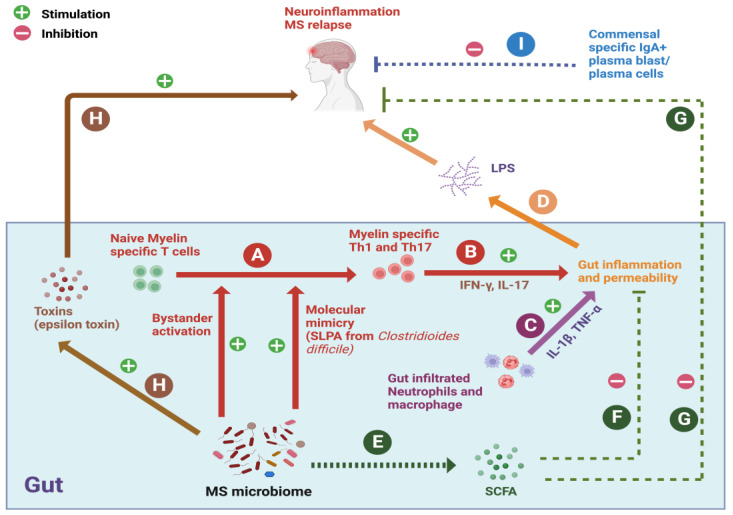
Role of gut–immune axis in MS pathogenesis. (**A**) Dysbiotic MS microbiota can promote the differentiation of myelin-specific T cells into Th1/Th17 cells by bystander action and molecular mimicry. (**B**,**C**) Myelin-specific Th1/Th17 cells, neutrophiles, and macrophages can cause gut inflammation and increased gut permeability by proinflammatory cytokines. (**D**) Increased gut permeability leads to the passage of bacterial endotoxins from the gut to periphery including the CNS, which may activate microglia and astrocytes to cause MS onset and/or progression. (**E**–**G**) MS gut microbiota is unable to produce sufficient levels of SCFAs, which may further promote intestinal permeability and neuroinflammation. (**H**) Certain bacterial species produce toxins and promote CNS pathology in MS. (**I**) Mobilization of IgA+ plasma blast and/or plasma cells from the gut to the CNS can significantly suppress neuroinflammation, and gut dysbiosis may suppress the development of IgA+ B cells in the gut. Schematic diagram was created using BioRender.com (accessed on: 13 September 2023).

**Figure 2 ijms-24-14756-f002:**
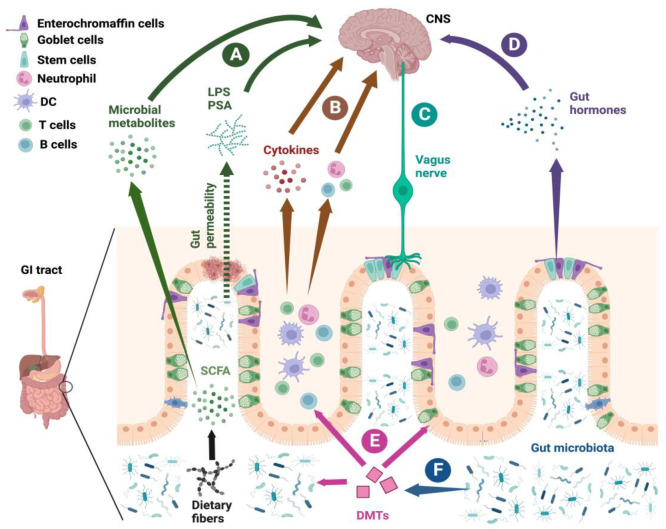
Mechanisms of gut–immune-axis-mediated effect on the CNS. (**A**) Microbial metabolites (SCFAs from dietary fibers and tryptophan metabolites) generated from gut microbiota and bacterial endogenous components (LPS and PSA) pass to the circulation due to leaky gut and affect peripheral immunity and glial cells in the CNS. (**B**) Migration of gut-resident innate and adaptive immune cell populations to the CNS and their cytokines can modulate MS pathogenesis. (**C**) The gut wall is innervated with both afferent and efferent fibers of the vagus nerve. Efferent nerve fibers carry impulses from the CNS to the gut and affect its physiology. On the other hand, afferent fibers carry impulses from the gut to the CNS. The activity of these fibers can be modulated by the diffusion of bacterial components (like LPS) and metabolites (like SCFAs) or hormonal signals from specialized EECs that are capable of sensing luminal bacterial content. (**D**) Gut-microbiota-derived LPS and SCFAs modulate the production and release of gut hormones (CCK, ghrelin, Peptide YY, GLP-1, 5-HT) from EECs. Gut hormones potentially suppress Th17 responses and neuroinflammation by attenuating activated microglia. (**E**) Disease-modifying therapies (DMTs) can affect gut immune cells and epithelial cells to induce changes in gut microbiota, which may contribute toward their therapeutic effects. By promoting anti-inflammatory gut bacteria, DMTs can enhance the development of Tregs or the production of regulatory cytokines in the gut and their circulation, which can suppress CNS autoimmunity. (**F**) Gut microbiota may modulate the response to DMTs. Schematic diagram was created using BioRender.com (accessed on: 13 September 2023).

**Table 1 ijms-24-14756-t001:** Effect of various gut bacteria on immune system in MS patients.

Bacterial Taxa	Level in MS	Functional Effect	Reference
*Clostridium (Clostridia cluster XIV* and *IV)*	Decreased	Decreased regulatory T cells (Treg) and IL10 production	[28]
*Prevotella*	Decreased	Differentiation of Th17 cells	[9]
*Streptococcus mitis* (*S. mitis*) and *Streptococcus oralis*	Increased		
Methanobrevibacter	Increased	Activation of T cells and monocytes	[18]
*Akkermansia*			
*Butyricimonas*	Decreased		
*Parabacteroides distasonis*	Decreased	Decreased anti-inflammatory IL-10-expressing human CD4+CD25+ T cells and IL-10+FoxP3+ Tregs in mice Increased differentiation of Th1 type cells and reduced proportion of CD25+FoxP3+ Treg cells	[12]
*Acinetobacter calcoaceticus*	Increased		
*Flavonifractor plautii*	Increased	Correlate positively with increased monocytes and neutrophils, and blood cell gene expression of IL17A and IL6	[29]
*Clostridium leptum*	Increased	Correlate positively with increased Type 1 IFN-induced blood cell genes: MX1, IFIT1, IFI44L, and IFI27	
*Flavonifractor*	Increased	Corelate positively with increased serum TNF-α	[30]
*Faecalibacterium* and *Roseburia*	Decreased	Corelate negatively with serum TNF-α	
*Faecalibacterium*	Decreased	Corelate positively with serum IL8 and MIP-1a	

**Table 2 ijms-24-14756-t002:** Important ongoing or completed studies addressing the effects of different diets on MS.

Diet/Intervention	Study Cohort or Design	Outcome	Clinical Trial/Ethical Committee Approval Number	Reference
High-fiber supplement (NBT-NM108)	RRMS	Ongoing study	NCT04574024	Not applicable
Propionate (PA)	RRMS	Significant increase in functionally competent Treg cells and decrease in Th1 and Th17 cells after two weeks. Reduced annual relapse rate, disability stabilization, and reduced brain atrophy after 3 years of PA intake.	15-5351 4493-12 17-6235 357_17B	[102]
High-vegetable/low-protein diet (HV/LP diet	RRMS	Induction of IL10-producing monocytes and Tregs. Increase in abundance of *Lachnospiraceae* family. Decrease in relapse rate and Expanded Disability Status Scale score.	Not available	[103]
Healthy/Mediterranean diet	Case control retrospective dietary recall studies	Lower risk of clinically isolated syndrome (CIS). Lower risk of MS.	Not applicable	[108,109]

**Table 3 ijms-24-14756-t003:** Effect of DMTs on gut microbiota composition in MS.

Type of DMT	DMTs	Microbial Changes		Reference
		Increase	Decrease	
Injectables	Interferons (IFN)	*Bacteroides uniformis* *Prevotella* genus *Sutterella* genus *Ruthenibacterium lactatiformans*	*Akkermansia muciniphila* *Sarcina* genus *Prevotella copri*	[10,11,16,18,131]
	Glatiramer Acetate (GA)	*Prevotella* genus *Sutterella* genus	*Bacteroides uniformis* *Lachnospiraceae* family *Veillonellaceae* family *Akkermansia muciniphila* *Sarcina* genus	[10,11,18,126]
Oral	Dimethyl fumarate (DMF)	*Lactobacillus pentosu* *Roseburia intestinalis* *Ruthenibacterium lactatiformans* *Bacteroidetes* phylum	*Bacteroides stercoris* *Clostridium* species *Eubacterium* species *Coprococcus eutactus* *Enterococcus gilvus* *Lachnospiraceae* families *Veillonellaceae* families *Firmicutes* phyla *Fusobacteria* phyla *Clostridiales* order	[10,16,126,132,133]
		Streptococcus, Haemophilus, Clostridium, Lachnospira, Blautia, Subdoligranulum, and Tenericutes in MS subjects with side effects	*Bacteroidetes, Barnesiella, Odoribacter, Akkermansia*, and some *Proteobacteria* families in MS subjects with side effects	
	Fingolimod	*Ruminococcaceae PAC001607* *Ruthenibacterium lactatiformans*	*Bacteroides finegoldii CAG:203* *Roseburia faecis* *Blautia* species	[10,16]
Infusion	Ocrelizumab (Anti-CD20)	*Faecalibacterium prausnitzii* *Ruthenibacterium lactatiformans*		[16]
	Natalizumab	*Phascolarctobacterium* sp. *CAG:207* *Ruminococcaceae PAC001607* *Ruthenibacterium lactatiformans*	*Bacteroides uniformis* *Prevotella* species *Bifidobacterium longum* *Akkermansia muciniphila*	[10,11,16]

## Data Availability

Not applicable.

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
