# Peer review of "Interaction of the Gut Microbiome and Immunity in Multiple Sclerosis: Impact of Diet and Immune Therapy"

_ijms, 2023, doi:10.3390/ijms241914756_

Round 1

Reviewer 1 Report

I believe that this is an interesting review on MS and gut immunity connection, together with factors (dietetic and pharmaceutical) that may affect both of these bidirectionallly.  

I propose the acceptance of this manuscript after some modifications.

Some issues are listed below.

1.     The authors analyze the effect of DMTs. However the title “Effects of Diet and Therapeutics on Gut Immunity in Multiple Sclerosis.” refers to therapeutics in general. Please rephrase.

2.     The article mainly describes the interrelation of gut microbiome and immunity with MS. This is not reflected in the title. The title emphasizes on the effects of diet and therapeutics. However, only a small part of the manuscript has to do with this topic.  

3.     Please adjust the quality and the size of figure 1. It is difficult to be analyzed by the reader.

4.     Please make a thorough revision of the text for potential grammar, punctuation, typographical or expression errors or omissions.

NA

Author Response

  1. The authors analyze the effect of DMTs. However, the title “Effects of Diet and Therapeutics on Gut Immunity in Multiple Sclerosis.” refers to therapeutics in general. Please rephrase.

Response: We changed the title to specify DMTs (see line 2-3).

  1. The article mainly describes the interrelation of gut microbiome and immunity with MS. This is not reflected in the title. The title emphasizes on the effects of diet and therapeutics. However, only a small part of the manuscript has to do with this topic.

Response: We changed the title to reflect the content of manuscript (see line 2-3).

  1. Please adjust the quality and the size of figure 1. It is difficult to be analyzed by the reader.

Response: We adjusted the quality and size of figure 1 (see line 255-256).

  1. Please make a thorough revision of the text for potential grammar, punctuation, typographical or expression errors or omissions.

Response: We made a through revision as suggested by reviewer. Minor corrections are highlighted in yellow throughout the manuscript.

Reviewer 2 Report

The article "Effects of Diet and Therapeutics on Gut Immunity in Multiple Sclerosis" will discuss the function of gut immunity throughout the pathophysiology of MS and how diet and DMTs can modify it via gut flora. According to specific data, gut immunity plays a big part in the immunopathogenesis of MS. The authors discuss how nutrition and DMT can modulate MS pathogenesis by influencing the gut microbial ecology, affecting gut immunity.

The manuscript is well-written, contains well-documented information, and can be accepted for publication after adding two figures.

1)    Graphical abstract

2)     A figure that combined data from the 2.1 through 2.4 parts.

Author Response

The manuscript is well-written, contains well-documented information, and can be accepted for publication after adding two figures.

  1. Graphical abstract

Response: We included two graphical abstracts in figure 1 and 2 (see line 67-79 and line 255-256).

  1. A figure that combined data from the 2.1 through 2.4 parts.

Response: We included figure 1 to depict parts 2.1 through 2.4 (see line 67-79).